# Exploration of a Flexible Metasurface for Strain Sensors: A Perspective from 2D Grating Fabrication to Spectral Characterization

Hao Hu [1,2] and Bayanheshig [1,*]

1 National Engineering Research Centre for Diffraction Gratings Manufacturing and Application, Changchun Institute of Optics, Fine Mechanics and Physics, Chinese Academy of Sciences, Changchun 130033, China
2 University of Chinese Academy of Sciences, Beijing 100049, China
* Correspondence: bayin888@sina.com

**Abstract:** The flexible plasmonic metasurface is a novel optical device consisting of a large number of subwavelength-sized noble metal (gold, silver, etc.) structures arranged in a specific pattern on a flexible substrate. The usual method for a fabricating flexible metasurface is to build nanostructures on rigid substrates and then transfer them to flexible substrates. However, problems such as structural distortion and structural loss can occur during fabrication. To address these issues, this work improved the process to fabricate and characterize a flexible plasma 2D grating–a type of metasurface composed of gold cubelets with a thickness of 50 nm and a side length of 250 nm. First, an electron beam lithography method modified by proximity effect correction was used to fabricate nanostructures on a rigid substrate. Then, the structures were transferred by a chemical functionalization and a sacrificial layer etching method. In addition, the feasibility of using flexible plasmonic 2D gratings as strain sensors was investigated in this work through a stretching test. Experimental results show that electron beam lithography improved by correcting the proximity effect enabled the fabrication of more precisely shaped nanostructures; the chemical functionalization method significantly improved the transfer yield; and the spectroscopic analysis in the stretching test demonstrated the potential of the flexible plasmonic 2D gratings for sensing applications.

**Keywords:** flexible metasurface; 2D-nano grating; electron beam lithography; proximity effect correction; gold functionalization





## 1. Introduction

Two-dimensional plasmonic gratings are a type of metasurface that has been widely used in fields such as wavelength identification and multi-channel orbital angular momentum generation [1,2]. The techniques for fabricating such metasurfaces on a flexible substrate have attracted more and more attention [3–6]. Compared to photonic devices on rigid substrates, flexible photonic devices break the mechanical limitations of rigid devices and have promising applications in emerging fields such as biotechnology [7], wearable devices [8], epidermal sensing [9] and optoelectronic devices [10]. The study of flexible plasmonic 2D gratings as one of the most fundamental and widely used flexible metasurfaces has not only explored their own properties, but also provided ideas for the design and fabrication of other flexible metasurfaces.

With the development of nanotechnology, numerous methods are now available for fabricating flexible optical devices, e.g., lithography [11], nanosphere lithography [12], shadow mask lithography (SML) [13], laser microlens array lithography [14], and direct laser writing (DLW) [15]. Compared to other techniques, electron beam lithography has the advantage of higher resolution and does not require masks during exposure [16]. The most commonly used method for fabricating flexible metasurfaces is to fabricate nanostructures on rigid substrates using electron beam lithography and then transfer them to flexible

substrates, as shown in Figure 1. Commonly used flexible substrates include Metaflex [17], PMMA [18], PET [19], and PDMS [20].

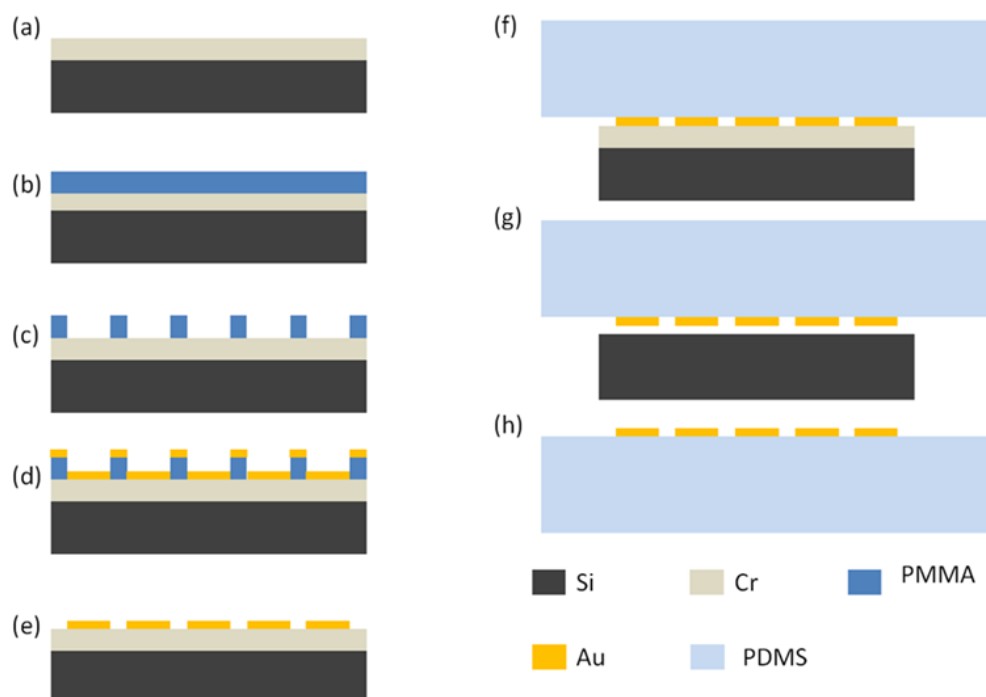

**Figure 1.** Flexible optical device fabrication flow chart: (**a**) chromium evaporated on substrate; (**b**) resist is spin coated on the chromium surface; (**c**) The designed structure is exposed on the substrate through lithography and development; (**d**) gold evaporation (**e**) lift off; (**f**) flexible substrate is pressed onto the structures; (**g**) Cr sacrificial layer etched off; (**h**) successful transfer of gold structures.

Initially, a chromium layer is deposited on the silicon substrate as a sacrificial layer. Then the resist (usually polymethyl methacrylate) is spin-coated on the chromium surface, and the designed structures are built by electron beam lithography (EBL) and metal deposition. The PDMS substrate is then pressed onto the structures. After etching away the sacrificial layer, the structures on the rigid substrate are attached to the flexible substrate by van der Waals forces. Finally, the transfer of the structure is completed by direct inversion or assisted transfer [21]. However, there are critical problems with the fabrication process described above. First, the electron beam has proximity effects, i.e., the scattering of electrons in the resist and the substrate causes a change in the exposure position, resulting in distortion of the structures. Second, the nanostructures cannot permanently adhere to the flexible substrate by van der Waals forces alone, thus structure loss and detachment constantly occur [22,23].

This work used a proximity effect correction method to accurately calculate the exposure dose required for each exposure point. This has greatly improved the accuracy of the structural shape without increasing the experimental complexity. In addition, we increased the transfer yield of the structures by chemically functionalizing the materials to be transferred, and subsequent experiments demonstrated that the structures did not detach during the stretching process. At last, stretching tests were performed to investigate the feasibility of using the flexible 2D grating as a strain sensor. The results show that flexible plasmonic 2D gratings have great potential for sensing applications. Compared to other flexible devices, the flexible plasmonic 2D gratings operate in the visible light region [24] and are more sensitive to strain [12].

## 2. Materials and Methods

### 2.1. Theory

There are two types of proximity correction, dimensional correction [25,26] and dose correction [27–29]. In dimensional correction, the underexposure or overexposure caused by proximity effects is compensated for by changing the size of the pattern itself or the surrounding patterns. Dose correction is performed by dividing the pattern into regions and then applying different exposure doses to each region to eliminate the uneven energy distribution caused by proximity effects. In this work, we have taken the dose correction approach and improved it. By specifying that the distance between each exposure point should be the same, the computational process is simplified.

In the transfer process, the materials to be transferred are chemically functionalized to increase the transfer yield. The purpose of functionalization is to increase the difference in adhesion between the materials. The functionalization of gold has been studied by many, and the interactions between thiol groups and gold have been demonstrated [30–32]. Gold binds very readily to thiols (Au-S) [33], which is one of the most stable bonds known and is often used in biosensors [34–37]. In this work, a chemical liquid deposition formulation of 3-mercaptopropyltriethoxysilane (MPTS, Thermo Fisher Scientific, MA, USA) SAMs [38] was used. Two different solutions were prepared, representing two stages of this sensitive functionalization process. The first stage is the growth of MPTS SAM on the gold structure. The second stage is the hydrolysis of the triethoxysilane [39]. The ethoxy group of MPTS is bonded to the hydroxyl group of PDMS through a condensation reaction, resulting in a tighter bond between PDMS and the gold structure.

#### 2.1.1. Proximity Effect Correction

Consider a set of points defined by their coordinates $(x_i, y_i)$, $i = 1, \ldots, N$. as shown in Figure 2a and the exposure dose at each spot is $D_i$. Due to the scattering of the electrons in the resist and the backscattering from the substrate, the dose is not only applied at the point, but also distributed in the surrounding area. The redistribution of the dose delivered to the resist by exposure at $(x_0, y_0)$ is described by the "point spread function" $PS(x, y, x_0, y_0)$. Typically, this can be defined as a double Gaussian function [40]:

$$PS_{DG}(r) = \frac{1}{\pi(1+\eta)} \left\{ \frac{1}{\alpha^2} \exp\left[-\frac{r^2}{\alpha^2}\right] + \frac{\eta}{\beta^2} \exp\left[-\frac{r^2}{\beta^2}\right] \right\} \tag{1}$$

In Equation (1), the first term represents the exposure dose delivered by the main beam with radius $\alpha$, and the second term describes the exposure dose delivered by backscattered electrons with a corresponding width $\beta$. The parameter $\eta$ represents the relative weight of the two exposure contributions. r is the distance from the point of incidence of the beam. The parameters $\alpha$, $\beta$, $\eta$ depend mainly on the substrate material and the beam energy.

Now, when all exposure doses are performed for N points, the ith point receives not only its own dose, but also a from its neighbors, and from itself is passed to the neighbors. The actual dose received at the ith point would be:

$$S_i = \sum_{j=1}^{N} PS(x_i, y_i, x_j, y_j) \cdot D_j = \sum_{j=1}^{N} PS_{DG}\left(\sqrt{(x_i - x_j)^2 + (y_i - y_j)^2}\right) \cdot D_j \tag{2}$$

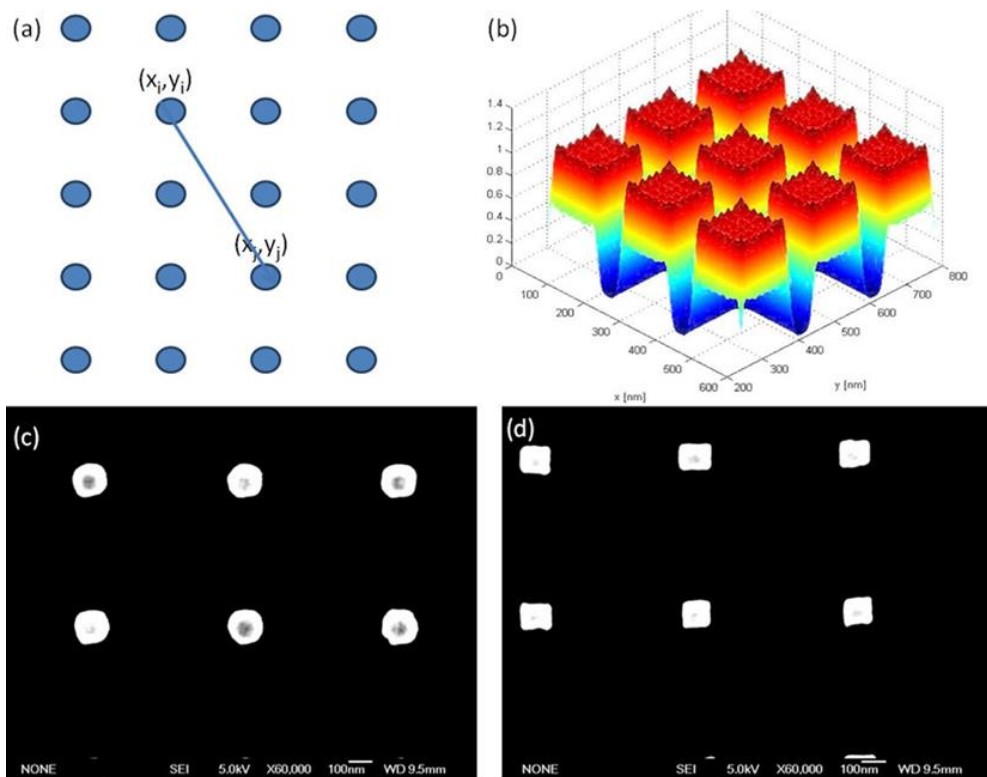

**Figure 2.** (**a**) Schematic diagram of the electron beam exposure points. (**b**) Schematic diagram of the exposure dose at each point in the 100 nm square region calculated by MATLAB (2016b, MathWorks, CA, USA). (**c**) Electron micrograph of a square 2D grating structure without proximity effect correction, with a side length of 100 nm. (**d**) Electron micrograph of a square 2D grating structure with proximity effect correction, with a side length of 100 nm.

If the distance $\delta$ of the points is equidistant and comparable to $\alpha \ll \beta$, we can improve the double Gaussian as follows:

$$PS_{ij} = \frac{1}{\pi(1+\eta)} \left\{ \frac{\pi}{2\delta^2} \left[ erf\left(\frac{x_i - x_j + \delta/2}{\alpha}\right) - erf\left(\frac{x_i - x_j - \delta/2}{\alpha}\right) \right] \right.$$
$$\cdot \left[ erf\left(\frac{y_i - y_j + \delta/2}{\alpha}\right) - erf\left(\frac{y_i - y_j - \delta/2}{\alpha}\right) \right]$$
$$\left. + \frac{\eta}{\beta^2} exp\left[ -\frac{(x_i - x_j)^2 + (y_i - y_j)^2}{\beta^2} \right] \right\} \tag{3}$$

Then the individual exposure dose values can be computed from:

$$D_i = \sum_{j=1}^{N} PS_{ij}^{-1} \tag{4}$$

The exposure dose of each point can be calculated using the MATLAB program, and the purpose of the calculation is to unify the energy distribution of the entire area to be exposed by adjusting the exposure dose. First, an initial exposure dose is set (usually given empirically) and then the energy distribution is calculated for all exposed patterns. Then, the dose is adjusted to the calculated energy distribution at each point and the energy distribution is calculated again. This distribution may be more homogeneous than before, but still insufficient, so the exposure dose at each point must be further adjusted and the energy distribution recalculated. The energy distribution is calculated by continuously adjusting the exposure dose at each point until the difference between the energy distribution and the ideal value is within an acceptable range, at which point the

calculation is terminated. In the calculation, the ideal energy distribution is usually set to 1 at each point. Figure 2b shows the results of exposure dose calculation for a square grating structure with a side length of 100 nm. It can be seen that the energy distribution in the square is basically uniform. In fact, each of the nine identical squares requires a unique exposure dose for each calculated exposure point. Figure 2c,d show the comparison of the structures with and without proximity effect correction, and the designed structures are both squares with a side length of 100 nm and a thickness of 50 nm.

### 2.1.2. Functionalization

The functionalization of the structure has two goals, one is to make the silicon and PDMS as non-adhesive as possible, and the other is to make the gold structure and PDMS as adhesive as possible. In order to reduce the adhesion between the silicon substrate and PDMS, we introduced hexamethyldisilazane (HMDS), the chemical expression is shown in the Figure 3b. Firstly, HMDS volatilizes to form a self-assembled monolayer on the silicon substrate, then, by heating, HMDS splits into two trimethylsilane (Si(Me)$_3$) molecules. This passivates the silicon surface by bonding the siloxane to the hydroxyl group of the native oxide on the silicon wafer, effectively reduced the adhesion of PDMS to the silicon surface. The second objective is to improve the adhesion of PDMS to gold structures, and the hydrolysis of triethoxysilanes is the key principle. The 3-mercaptopropyltriethoxysilane (MPTS) contains a thiol functional group (-SH) that easily binds to gold nanostructures and can be used as a linking molecule between gold and PDMS, as shown in Figure 3a. The ethoxy group of MPTS is replaced by a hydroxyl group through the S$_N$2 reaction. In order to bind the hydroxyl group of PDMS by condensation reaction, chloroform is introduced as a catalyst, here HCCl$_3$ is the electron donor which catalyzes the production of a OH$^-$ nucleophilic reagent by heterolytic fission of H$_2$O.

(a)

Hexamethyldisilazane (HMDS)

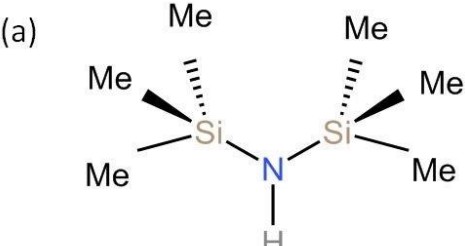

(b)

3-Mercaptopropyltriethoxysilane (MPTS)

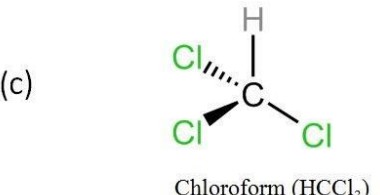

(c)

Chloroform (HCCl$_3$)

**Figure 3.** Structural formula of the chemicals required for the functionalization process; (**a**) 3-mercaptopropyltriethoxysilane (MPTS); (**b**) hexamethyldisilazane (HMDS) (**c**) chloroform (HCCl$_3$).

2.1.3. Optical Measurement

When light illuminates the surface of nanoparticles of a noble metal, the free electrons near the surface of the nanoparticles oscillate collectively. When the frequency of the incident photon matches the oscillation frequency, the metal nanoparticles produce a strong absorption of photon energy, i.e., local surface plasmon resonance (LSPR) occurs. Additionally, a strong absorption or scattering peak appears in the spectrum. The frequency of electron oscillation is related to the material, size and shape of nanoparticles. For a single nanoparticle, the irregular shape leads to multiple electronic oscillation modes, which are reflected in the spectrum as multiple absorption or scattering peaks. In contrast, the periodic arrangement of many identical nanoparticles interacting together with the incident light leads to a new mode. This is called the surface lattice resonance (SLR) mode or grating mode. The grating mode can theoretically be changed by altering the shape or period of the nanoparticles. Stretching the flexible substrate can change the period of the 2D grating, which in turn changes the position of the resonance peak of its dark-field scattering spectrum. For example, if the sample is stretched longitudinally and the ratio between the stretched length and the original length is 0.5, the period of the grating becomes 1.5 times the original period. According to the grating equation, the lattice mode should exhibit a redshift as the period increases. Consequently, small strains can be detected by observing the spectrum.

*2.2. Fabrication and Characterization*

The wafer was first dipped in acetone and ultrasonically cleaned for 5 min, then dipped in isopropanol and ultrasonically cleaned for another 5 min. The wafer was then rinsed with deionized water for 30 s and blown dry with nitrogen. The cleaned wafer was then placed in an evaporator (Pfeiffer Vacuum PLS 570, Aßlar, Germany), and 200 nm Cr was evaporated onto the silicon surface by electron beam evaporation, used as a conductive layer for electron beam lithography and a sacrificial layer for pattern transfer. The chromium-plated wafers were placed on the spin-coater and a 2.5% MIBK solution of PMMA was dropped onto the top surface and spun at 2600 rpm for 6 s to ensure that the PMMA was evenly distributed on the sample surface, and then spun at 5000 rpm for 60 s to achieve the desired thickness of about 160 nm. The exposure was performed by a conventional electron beam lithography method using a JEOL JSM-6500F field-emission gun scanning electron microscope combined with a XENOS pattern generator. The unique feature of the EBL in this work is that the pattern file used for the exposure was improved by correcting for the proximity effect. After exposure, the sample was developed in a 1:3 solution of MIBK and isopropanol for 75 s, which removed the exposed part and preserved the unexposed part so that the designed pattern could be reproduced on the substrate. Then the developed substrate was placed in the evaporator, and 50 nm of Au was evaporated onto the sample by thermal evaporation. Finally, the sample was immersed in acetone for more than 6 h, and the unexposed photoresist was washed off the sample together with the Au, leaving the exposed part with the gold structure. In this way, the structures were fabricated on the rigid substrate. The next step was to transfer the gold structure on the rigid substrate to the PDMS mold. Before preparing the PDMS substrate, the mold must be hydrophobized. The crucible was placed in a desiccator, then 3 mL of HMDS was poured into the crucible and the mold was placed on the crucible. The desiccator was then pumped to a pressure below 100 mTorr and allowed to rest for 2 h. During this time, the HMDS evaporated and formed a self-assembled monolayer on the sample and mold. The sample was then baked at 150 °C for 30 min to complete the hydrophobic treatment.

Next, the preparation of the PDMS substrate was started. First, 10 parts of PDMS pre-elastomer was mixed with 1 part of hardener (Dow Corning SYLGARD 184 was used in this work) and stirred for 5 min to produce a homogeneous mixture. Then the mixed PDMS was placed in a vacuum chamber for 30 min to remove all air bubbles. The hydrophobic mold was placed on the bench and the clear PDMS was carefully poured onto the mold and cured in an oven at 150 °C for 10 min. After demolding, the desired PDMS substrate was

obtained. The mold functionalized with HMDS allows the PDMS substrate to be demolded in an intact state.

Normally, the transfer requires functionalization of the silicon substrate. However, in this case, functionalization of the silicon is not required because the surface layer is present. Therefore, HDMS was used only to hydrophobize the silicon mold. 5 mL of a 5 *v/v*% MPTS (Thermo Scientific™) tetrahydrofuran (THF) solution was prepared in an airtight bottle. The sample with the prepared structures was immersed in the MPTS solution for more than 1 h. Then the sample was taken out, rinsed twice with fresh tetrahydrofuran solution, and then blown dry with nitrogen. The sample was first held over a heater of 80 °C MilliQ water for a fraction of a second to create a thin layer of water mist on the sample surface. Once the water mist disappeared, the sample was immersed in the MPTS solution for 3 min. The sample was then removed from the solution, rinsed twice with fresh THF, and dried with nitrogen. This completed the functionalization of the gold structure, and the transfer of the gold structure began next. The flexible PDMS substrate was placed on the previously prepared nanostructure and firmly pressed with a fixture. The PDMS adhered well to the nanostructures and the chromium sacrificial layer. The fixture and sample were then placed in TechniStrip Cr01 chromium etching solution and rested. In order for the sacrificial layer to be fully etched, the etching time was one week. The etching solution enters between the silicon substrate and the PDMS, and when the sacrificial layer is dissolved, the structures are released and adheres to the PDMS substrate. The TechniStrip Cr01 etching solution is highly selective and does not attack PDMS and gold during Cr etching [41]. Once etching was complete, the transfer step was terminated. The PDMS substrate was rinsed with water and dried with nitrogen. PDMS (cured at 150 °C) is elastic and deforms linearly up to 50% strain. In Figure 4c,d, the SEM image shows the transferred nanostructure assembly. Typical transfer yields are from 95% to 99%.

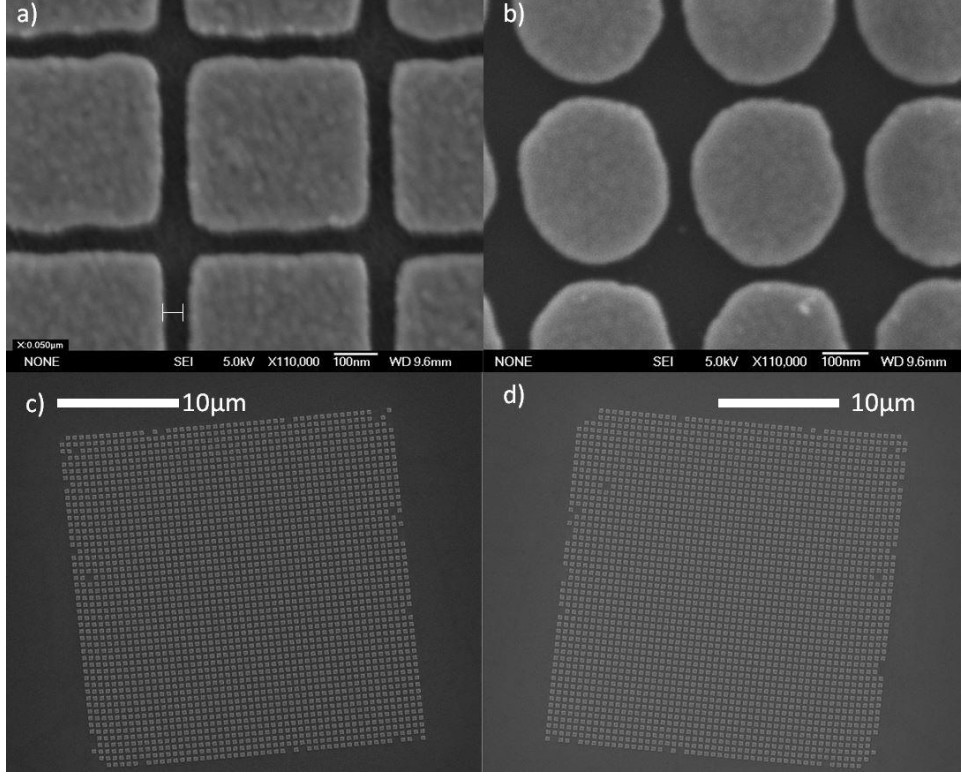

**Figure 4.** (**a**) SEM images of a 2D gold grating on a sacrificial layer of Cr with proximity effect correction; (**b**) SEM images of a 2D gold grating on Cr without proximity effect correction; (**c**) SEM image of a 2D gold grating with edge length of 200 nm and period of 500 nm before structural transfer; (**d**) SEM image of the grating structure transferred to PDMS substrate in (**c**).

Dark-field characterization was performed using a Zeiss Axio Scope A1 reflectance setup. The incident light was unpolarized and produced by a 100 W halogen lamp focused through a dark-field spot lens with a numerical aperture of 0.95 to 0.99. A 100× objective with a numerical aperture of 0.9 was chosen to collect the scattered light. The collected light was analyzed using a grating spectrometer (LOT SR -303i-B). The grating had a linear density of 150 L/mm and a central wavelength of 700 nm. The detector used was a camera (Andor iDus CCD) with a pixel size of $15 \times 15$ μm$^2$ and a $2000 \times 256$-pixel chip was used as a detector. The slit width was 100 μm, so a rectangle of $1 \times 1.05$ μm$^2$ was used for spectral analysis. The two opposite sides of the flexible substrate were fixed to the stretching device; see Figure 4a,b. The initial length between the two fixtures was set to 10 mm, and each stretching length was 1 mm. The exposure time was 3 s, and the measurements were repeated five times for averaging. The spectra of the structures to be measured, the structureless substrate, the light source and the dark current were measured separately and calculated using Equation (5). The result obtained was the corrected scattering spectrum.

$$Scattering = \frac{I_{str} - I_{background}}{I_{lamp} - I_{dark}} \tag{5}$$

where $I_{str}$ is the light intensity scattered by the nanostructure, $I_{background}$ is the light intensity scattered by the substrate. $I_{lamp}$ denotes the incident light intensity and $I_{dark}$ represents the scattered light intensity due to the dark-field current.

## 3. Results

A 2D flexible grating with more precise structures was produced by calculations to correct the proximity effect. As shown in Figure 4. Figure 4a,b are SEM images of the fabricated gold 2D grating structures with and without proximity effect correction, respectively. The structures in both figures are square and have a side length of 300 nm. It can be seen that the structures exposed with proximity effect correction have clearer boundaries and sharper corners, while the structures not corrected for proximity effects are significantly smaller and have rounder corners. Figure 4c shows the SEM image of the gold structures on a rigid substrate, and Figure 4d shows the gold structure on a PDMS substrate after transfer. The structures are cubelets with a 250 nm side length, a 50 nm thickness, and a period of 500 nm. It can be seen that almost all functionalized gold structures are transferred onto PDMS, with a transfer rate of over 99%.

In order to verify the feasibility of the prepared 2D plasmonic grating for applications in mechanical sensing, stretching experiments were also carried out in this study. As the stretching changes the period of the grating, it leads to a change in the peak position of its scattering spectrum. It is worth noting that the scattering peak caused by LSPR does not shift significantly because the shape of the gold structures is relatively stable during the stretching process. Theoretically, only the peak of the grating mode shifts during stretching. If the spectral resonance peaks shift significantly and linearly to the stretched length, the sensing effect has been shown to be good.

The stretching experiment is shown in Figure 5. Figure 5a is a photograph of the PDMS substrate. Figure 5b shows the homemade stretching equipment, and the length of stretching can be controlled by rotating the spiral micrometer on the right side. At the same time, the flat bottom can keep the sample stable during the stretching process and facilitate the measurement of the spectrum. Figure 5c is a dark-field microscope image of the 2D flexible grating in the unstretched state. Figure 5d is a dark-field microscopic image of the 2D flexible grating after stretching by 50%. Each bright square in the figure is a set of 2D gratings. Usually, more than one set of gratings was fabricated and transferred to ensure the success of the experiment.

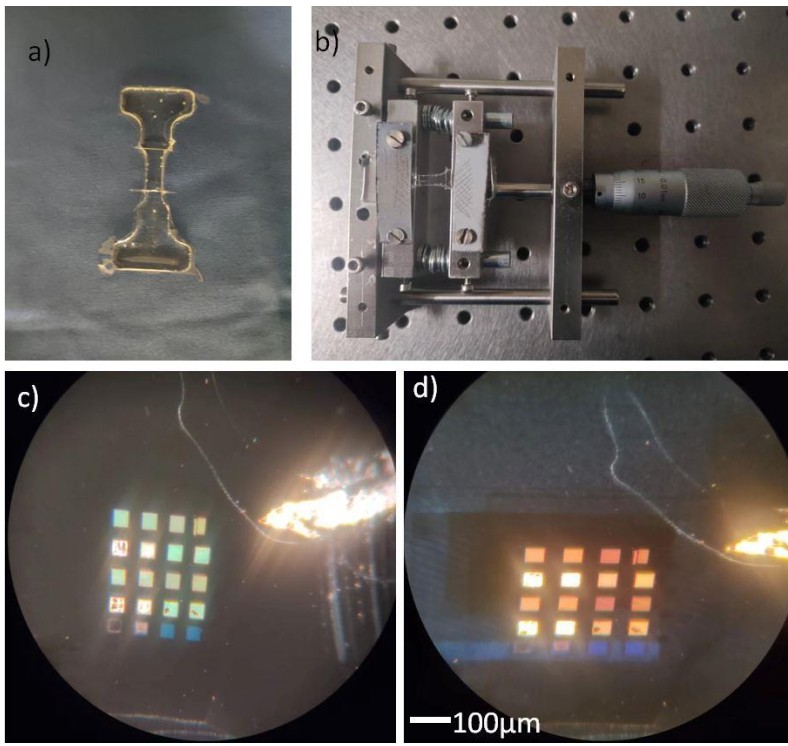

**Figure 5.** (**a**) photo of the PDMS mold; (**b**) homemade stretching equipment; (**c**) dark-field microscope image of unstretched flexible 2D grating; (**d**) dark-field microscope image of flexible 2D grating with 50% stretch length.

The scattering spectra measured during the stretching process are shown in Figure 6. Figure 6a shows a stacked plot of the normalized scattering spectra of a flexible plasmonic grating with a square edge length of 250 nm, a thickness of 50 nm, and a period of 500 nm. It should be noted that the spectrum in Figure 6a was not obtained from direct measurements. Instead, it was obtained by calibration of four sets of data, as shown in Equation (5). The values of its y-axis have only relative significance, showing the resonance position rather than the reflectance. The plasmonic resonance peak at a wavelength of about 850 nm remained essentially unchanged during the stretching process, while its grating mode showed a significant red shift with the change of the grating period. According to the empirical analysis [42], the leftmost scattering peak, at about 530 nm, is the vertical mode of the gold thin cubelet structure, i.e., the resonance peak resulting from the electron oscillation in the direction of cubelet thickness. As the PDMS substrate was stretched, no significant shift of the peak occurred. This indicates that the thickness of the gold cubelets did not change significantly with stretching. The rightmost peak, at about 860 nm, is the plasmon resonance peak of the gold structure, with the electron oscillation direction parallel to the stretching plane. This peak also exhibited almost no shift during the stretching process, indicating that the shape of the gold rogressedre did not change significantly in the direction parallel to stretching. The middle peak is the grating mode, which shifted from 570 nm to about 720 nm as the stretching progressed. This is due to the fact that as the sample is stretched, the period of the grating becomes larger, and the grating mode is red-shifted. Figure 6b shows the peak position of the spectrum obtained by Lorentz fitting. It can be clearly seen that peaks 1 and 3 remain stable as the stretching progresses, while peak 3 shifts significantly. By analyzing the change of spectral peak position with the stretching length, as shown in Figure 6b, the spectral shift was basically linear with the stretching length, taking into account the errors due to the lack of precision of the stretching equipment. The work at this stage did not test the stretching limit of the samples, but no significant structural detachment was found in the several stretching experiments that were performed. As shown in Figure 7 the dark-field microscope image of the flexible grating with the original

length restored after stretching. Compared with Figure 5c, the shape and color of the gratings did not change significantly. Figure 7b shows the comparison of the scattering spectra before and after stretching. It can be seen that the spectra before and after stretching did not show significant shifts. This indicated that there was no significant deformation and structure detachment of the sample during stretching.

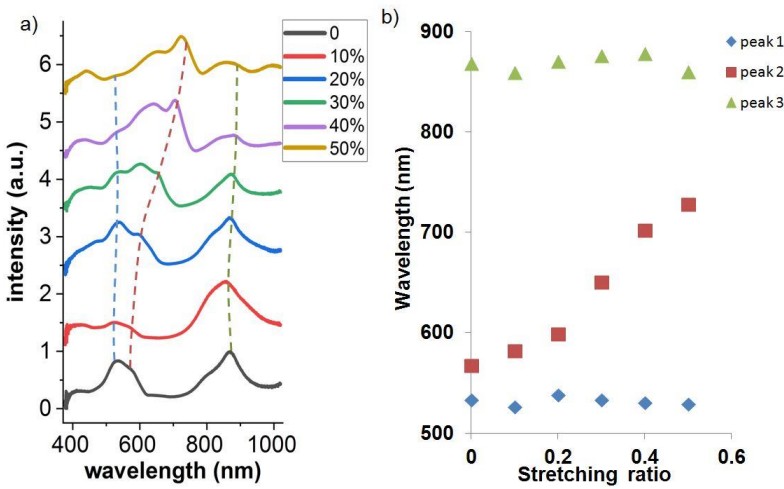

**Figure 6.** (**a**) Normalized scattering spectra of the 2D gold grating at different stretching lengths; (**b**) the resonance peak varies with the position of the stretching ratio.

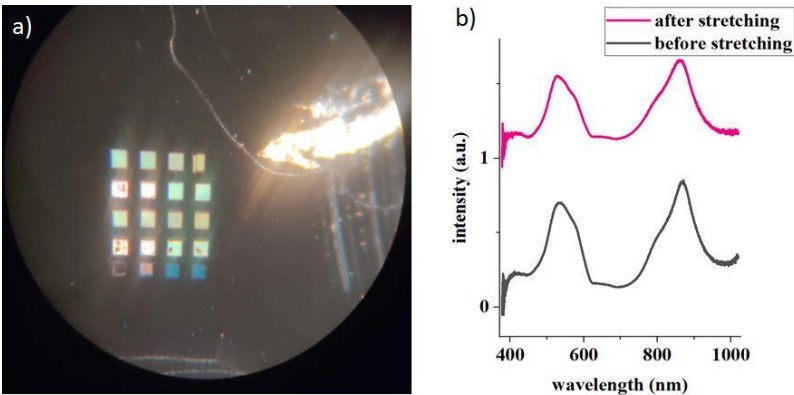

**Figure 7.** (**a**) dark-field microscope image of flexible 2D grating after stretching; (**b**) normalized scattering spectra of the flexible 2D gold grating before and after stretching.

## 4. Conclusions

In this article, we improved the electron beam lithography process by a double Gaussian function proximity effect correction. By precisely controlling the exposure dose, thin cubelet structures with straight edges and sharp corners were fabricated. In addition, in the transfer process, this study proposed a method to functionalize the gold structure and silicon mold using MPTS and HMDS. The adhesion between the gold structure and PDMS material was improved, while the adhesion between silicon and PDMS material was reduced. This resulted in a transfer yield of more than 99% and no significant structural detachment was found in stretching experiments. After fabrication, the feasibility of 2D flexible plasmonic gratings for strain sensing was also explored. By measuring the scattering spectra during the stretching process, we found that its resonance peak position shifted by 150 nm with a stretching ratio from 0 to 50%. Furthermore, the shift appeared to be obvious and stable, showing a good prospect for sensing applications. Meanwhile, there is still room for improvement in the current work. For example, different etching methods can be tried

to find a faster way to transfer the structures. In addition, well-established numerical simulation models can enable the optical properties of 2D gratings to be better studied.

**Author Contributions:** Conceptualization: Bayanheshig, H.H.; methodology: H.H.; software: H.H.; validation: Bayanheshig, H.H.; resources: Bayanheshig; All authors have read and agreed to the published version of the manuscript.

**Funding:** This research was funded by National Natural Science Foundation of China (Grant No. U21A20509).

**Conflicts of Interest:** The authors declare no conflict of interest.

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
