# Peer review of "Exploration of a Flexible Metasurface for Strain Sensors: A Perspective from 2D Grating Fabrication to Spectral Characterization"

_applsci, doi:10.3390/app121910007_

Round 1

Reviewer 1 Report

The authors demonstrate a method to calculate exposure dose by double Gaussian function when taking proximity effect correction into consideration. This method was applied in fabricating gold gratings, and the grating could be transferred to a flexible substrate with a high transfer yield and good sensing properties. I’m glad to recommend this paper if the authors can solve my concern below:

1.        The recyclability is a significant property for the flexible devices, so can you give the reflection spectra after stretching or more straining process?

2.        According to the grating equation, the lattice mode will redshift with the period increasing, can you explain the trend of the peak with stretching in Fig 6.a)? How does the period change with stretching proceeds?

Author Response

  1. The recyclability is a significant property for the flexible devices, so can you give the reflection spectra after stretching or more straining process?

A: The dark-field microscopy images after stretching and the comparison of spectra before and after stretching have been given in Figure 7.

  1. According to the grating equation, the lattice mode will redshift with the period increasing, can you explain the trend of the peak with stretching in Fig 6.a)? How does the period change with stretching proceeds?

A: The period change with the stretching has been explained in the “2.1.3 optical measurement” part.

Reviewer 2 Report

This article is very significant in the production of flexible two-dimensional plasmonic grating, according to the proximity effect correction defined by the double Gaussian function, the exposure dose for each point be calculated, based on this principle, the exposure dose of the 2D grating using electron beam lithography is controlled, and the gold structure and silicon mold is functionalized also, more than 99% in a transfer yield is achieved.

      Meanwhile, some suggestion needs to be considered:

First: in the 2D grating experiment manufacture process, by precisely controlling the exposure dose, clear boundaries and sharp corners are fabricated, and the adhesion between the gold structure and PDMS material is improved, and the adhesion between silicon and PDMS material is reduced, this conclusion is achieved, the measurement data should be gived in the article, or give the comparison explanation.

 Second: the image used in the figure 4(a) and 4(b), the grating with more precise structures has been produced by proximity effect correction, the same magnification should be showed if possible, in order to show the details especially in the corner shape and the boundaries.

Author Response

First: in the 2D grating experiment manufacture process, by precisely controlling the exposure dose, clear boundaries and sharp corners are fabricated, and the adhesion between the gold structure and PDMS material is improved, and the adhesion between silicon and PDMS material is reduced, this conclusion is achieved, the measurement data should be gived in the article, or give the comparison explanation.

A: Regarding the control of the exposure dose to produce a more angular structure, a stronger comparison has been given in Figure 4. Regarding the quantitative analysis of the contribution of chemical functionalization, there is no good way to measure it at the moment. However, the comparison before and after transfer given in Figs. 4c and 4d shows that a total of over 2500 metal square structures before transfer. After the transfer only 3 structures are lost, and the transfer yield is about 99.8%

Second: the image used in the figure 4(a) and 4(b), the grating with more precise structures has been produced by proximity effect correction, the same magnification should be showed if possible, in order to show the details especially in the corner shape and the boundaries.

A: This is indeed an important comment. However, the sample of the original image has been used as a PDMS transfer experiment and can no longer be observed by SEM. But we have re-made a new sample that provides a stronger contrast at the same magnification, which has been modified in Figure 4.

Reviewer 3 Report

The manuscript “Fabrication and characterization of flexible two-dimensional
plasmonic gratings
” is a very interesting work. In this work
an electron beam exposure procedure modified by proximity effect correction will be used with the aim of producing 2D gratings with clear edges and sharp corners. Moreover, the structures will be transferred by chemical functionalization and sacrificial layer etching method, which can reduce the loss rate of the structures significantly. Experiment results validate the importance of proximity effect correction in electron beam lithography and the necessity of chemical functionalization in gold nanostructure transfer. The results are consistent with the data and figures presented in the manuscript. While I believe this topic is of great interest to our readers, I think it needs major revision before it is ready for publication. So, I recommend this manuscript for publication with major revisions.

1. In this manuscript, the authors did not explain the importance of flexible two-dimensional materials in the introduction part. The authors should explain the importance of flexible two-dimensional materials.

2) Title: The title of the manuscript is not impressive. It should be modified or rewritten it.

3) Correct the following statement “The structures on rigid substrate are released by etching the sacrificial layer”.

4) Keywords: the flexible is missing in the keywords. So, modify the keywords.

5) Introduction part is not impressive. The references cited are very old. So, Improve it with some latest literature like 10.1515/epoly-2016-0154, 10.1016/j.cej.2020.126827,

6) The authors should explain the following statement with recent references, “The advantage of dimensional correction is that the scanning speed does not need to be changed during the exposure process, allowing it to maintain a high exposure efficiency; the disadvantage is that it does not meet the optimal exposure requirements for each exposure point and the accuracy is relatively low”.

7) Add space between magnitude and unit. For example, in synthesis “21.96g” should be 21.96 g. Make the corrections throughout the manuscript regarding values and units.

8) The author should provide reason about this statement “The middle peak is the lattice mode, which can be seen to move from 570 nm to about 720 nm as the stretching proceeds”.

9. Comparison of the present results with other similar findings in the literature should be discussed in more detail. This is necessary in order to place this work together with other work in the field and to give more credibility to the present results.

10) Conclusion part is very long. Make it brief and improve by adding the results of your studies.

11) There are many grammatic mistakes. Improve the English grammar of the manuscript.

Author Response

1) In this manuscript, the authors did not explain the importance of flexible two-dimensional materials in the introduction part. The authors should explain the importance of flexible two-dimensional materials.

A: This comment has been improved in the introduction section.

2) Title: The title of the manuscript is not impressive. It should be modified or rewritten it.

A: Very good suggestion, the title has been changed to “Exploration of flexible metasurface for mechanical sensor: a perspective from 2D grating fabrication to spectral characterization”

3) Correct the following statement “The structures on rigid substrate are released by etching the sacrificial layer”.

A: This comment has been improved in the introduction section.

4) Keywords: the flexible is missing in the keywords. So, modify the keywords

A: Has modified the keywords.

5) Introduction part is not impressive. The references cited are very old. So, Improve it with some latest literature like 10.1515/epoly-2016-0154, 10.1016/j.cej.2020.126827,

A: These two papers have been listed as references 18 and 10

6) The authors should explain the following statement with recent references, “The advantage of dimensional correction is that the scanning speed does not need to be changed during the exposure process, allowing it to maintain a high exposure efficiency; the disadvantage is that it does not meet the optimal exposure requirements for each exposure point and the accuracy is relatively low”.

A: This is a good suggestion, but after assessing the overall situation of the article, we decided to delete this statement to streamline the description of the proximity effect correction.

7) Add space between magnitude and unit. For example, in synthesis “21.96g” should be 21.96 g. Make the corrections throughout the manuscript regarding values and units.

A: Corrections have been made to the units in the text

8) The author should provide reason about this statement “The middle peak is the lattice mode, which can be seen to move from 570 nm to about 720 nm as the stretching proceeds”.

A: A description of the principle has been added in section 2.1.3.

  1. Comparison of the present results with other similar findings in the literature should be discussed in more detail. This is necessary in order to place this work together with other work in the field and to give more credibility to the present results.

A: This is a very good suggestion and has been improved at the end of the introduction section. And two new references have been introduced for comparison

10) Conclusion part is very long. Make it brief and improve by adding the results of your studies.

A: The conclusion section has been rewritten

11) There are many grammatic mistakes. Improve the English grammar of the manuscript.

A: Thank you for your kind comments, have made language adjustments throughout the text.

Reviewer 4 Report

This experimental contribution is dedicated to the method of fabrication of two-dimensional subwavelength plasmonic gratings (2D plasmonic metasurfaces) on stretchable substrates. To this purpose the authors used thin gold cuboids arranged with a square unit cell on a stretchable PDMS substrate. First the 2D cuboid arrays have been fabricated by e-beam evaporation on a chromium sacrificial layer deposited on a silicon substrate and then transferred to stretchable substrate by pressing the two structures against each other and then removing the sacrificial layer. The proximity effects problems that cause electron scattering have been corrected by using Double Gaussian optimization method. Surfaces were functionalized to ensure a good adhesion between the transferred nano-cuboids and the stretchable substrate.

Scientifically, the manuscript is sound and correct. It is mostly well organized and its English is quite good, with only a small number of minor mistakes. There are, however, some omissions, insufficient information on some points and even questionable parts that should be dealt with. After their correction, the manuscript will be acceptable for publication.

Please find below some point-by-point suggestions for the manuscript improvement.

1. Abstract should be rewritten to include all the necessary information (short description of the plasmonic metasurface geometry, a sentence mentioning the concrete materials and quantitative data (structure dimensions). An abstract should be the paper in little, succinctly stressing all the main points and including the wider context, the scientific motivation for solving the chosen problem, the novelty of the paper, the description of the methodology, the main conclusions, the applicability of the approach and the directions of future work. Many of these points are missing from present Abstract.

2. Please remove all use of future tense from Abstract and write it in the customary manner.

3. State of the art on the fabrication of plasmonic nanogratings should be better written. An example of well written state of the art is the authors’ own first paragraph of Introduction describing diffraction gratings generally. In contrast to this, the authors only presented a single approach to the fabrication of stretchable plasmonic nanogratings and their proposals to improve it. They skipped all alternative approaches, especially the emerging ones, and omitted to stress the scientific novelty of their method over the competing ones. Both e-beam intensity optimization and the grating transfer to stretchable substrates have been done many times before in many ways by other teams. The correction could be done by writing a paragraph between the one presenting the state of the art of gratings and the one describing what the authors claim represents the most widely used transfer method. Also, this claim merits at least one new reference and preferably a few.

4. What are its advantages over the other proposed strategies and which problem does your method solve that has been unsolved until now (by ANY transfer technique, no just yours)? Describe the novelty of your approach compared to all previous solutions, including yours.

5. Please write what is the Au thickness in 2.1.1. From Fig. 2b it would appear that it is about 1 nm, however no label with units is given with the z-axis, so it is just my guess because the remaining two axes’ units are nm.

6. The whole grating transfer procedure that you described appears overly complicated and extremely slow. Did you consider an approach that would include the most part of the processing you described, including depositing the gold islands onto a thin polished Si substrate and using a similar functionalization and transfer procedure to the one you already described to "glue" the islands to a new PDMS substrate? The difference is that you could try to etch away the complete silicon substrate in e.g. KOH. It would have lasted much shorter than the proposed removal of sacrificial chromium. In addition, the non-use of chromium would help in decreasing binding between gold cuboids and silicon.

7. The description of coupling between the propagating light beam and the localized surface plasmon polariton as written in lines 157-8 is unclearly written, because it could be understood that a propagating wave could couple with a flat conductive surface without any coupling structure, which is of course impossible. Coupling in your case occurs via diffraction on nanoparticles (and via grating coupling) which adds the necessary additional wavevector. You could rephrase your sentence like “When light illuminates the surface of nanoparticles made from noble metal (or generally from any plasmonic material), the free electrons close to the nanoparticle surface will collectively oscillate.”

8. In Section 4 a comparison of optical images of the structure in Figs 5c and 5d clearly shows that the shape of the flat golden cubelets is elongated along the horizontal axis when the substrate is stretched. Yet, when describing the results in Fig. 6 you comment that there is “no significant change in the shape of the gold structure in the direction parallel to the stretching”. Also you claim that the “vertical length” (is it the thickness or the lateral dimension normal to the stretching direction?) of “cubes” (you do not use any cubes but flat square islands that are best described as “thin cubelets”) does not significantly change during stretching. How is that even possible? Upon stretching, the dimensions along the unstretched axes should be shortened proportionally to the elongation to keep the volume constant. If your explanation were that the shape of the cubelets did not change upon stretching, it would contradict Fig. d. Besides, this would create a huge stress between the stretched substrate and the cubelets, with a result that the interfacial forces would be so large that at least some of the cubelets would simply detach. Kindly elucidate.

9. Do repeated stretching/releasing cycles cause detachment/falling off of the gold squares?

10. I had several issues with Fig. 6a. The ordinate axis is labeled by “intensity”. Why is this "Intensity" and not "reflectance"? Why did you use arbitrary units instead of the absolute ones and what do they actually represent? Also, I was unable to distinguish the peak at approx. 530 nm, especially at higher levels of stress, while at the bottommost point I could not discern it from the middle peak. Thus the positions of the dashed lines appear to me somewhat arbitrarily drawn. Your kind explanation would be most welcome here.

11. In the same figure, something is wrong with the color code of the dashed lines. Their order is different between the spectral dependences and stretching percentages. I would expect them to be in identical order.

12. Conclusion seems incomplete to me. What are the prospects for your further work? What are the possibilities of mass fabrication (which are critical for any practical use of your stretchable gratings), especially from the point of view of the fact that the sacrificial chromium removal lasts impractically long? What are the envisioned applications of the structures?

13. As mentioned above, English in the manuscript is rather good. However, that is not to say that it does not need corrections. In the following few points, kindly find my suggestions for language and style improvements.

14. Please, re-read your manuscript once again and correct the errors like “photonic devices is”, “2.1 theroy”, “in 150 °C”, “under nitrogen”, etc. What is “figure of breath”? “Desired level of PMMA thickness” should be “The desired PMMA thickness.” “The exposure part” can be simply “Exposure.” The sentence “Place 3 ml of HMDS into the crucible in the desiccator and place the holder with the mold on top of the crucible” is unclear, please rephrase it. I believe “Fix both sides of the flexible substrate  on  the stretching device” should be “Fix two opposing sides of the flexible substrate  to  the stretching device”

15. Take special care that the tenses throughout your paper (or at least throughout a single paragraph) match. You cannot start a paragraph in one tense, switch the next sentence to another, then continue in third, it is considered to be bad style. Especially take care that you sometimes needlessly used future tense (e.g. lines 17, 18, or 90) or imperative.

16. Please don’t use imperative at all (for instance, remove it from the lines 173-177) because you make your procedure description sound like a manual. Use past tense instead, like in the rest of that paragraph.

17. Remove “Track Changes” (red font) from line 103.

18. References are well chosen, most of them are quite recent and appropriate. However, it appears to be necessary to add new ones in Introduction, related to the state of the art of the fabrication of nanooptical elements on stretchable substrates.

19. In all references the doi numbers are missing.

Author Response

  1. Abstract should be rewritten to include all the necessary information (short description of the plasmonic metasurface geometry, a sentence mentioning the concrete materials and quantitative data (structure dimensions). An abstract should be the paper in little, succinctly stressing all the main points and including the wider context, the scientific motivation for solving the chosen problem, the novelty of the paper, the description of the methodology, the main conclusions, the applicability of the approach and the directions of future work. Many of these points are missing from present Abstract.

A: The abstract section has been rewriten

  1. Please remove all use of future tense from Abstract and write it in the customary manner.

A: This point has been revised in the current Abstract.

  1. State of the art on the fabrication of plasmonic nanogratings should be better written. An example of well written state of the art is the authors’ own first paragraph of Introduction describing diffraction gratings generally. In contrast to this, the authors only presented a single approach to the fabrication of stretchable plasmonic nanogratings and their proposals to improve it. They skipped all alternative approaches, especially the emerging ones, and omitted to stress the scientific novelty of their method over the competing ones. Both e-beam intensity optimization and the grating transfer to stretchable substrates have been done many times before in many ways by other teams. The correction could be done by writing a paragraph between the one presenting the state of the art of gratings and the one describing what the authors claim represents the most widely used transfer method. Also, this claim merits at least one new reference and preferably a few.

A: These suggestions have been adjusted in the introduction section.

  1. What are its advantages over the other proposed strategies and which problem does your method solve that has been unsolved until now (by ANY transfer technique, no just yours)? Describe the novelty of your approach compared to all previous solutions, including yours.

A: The introduction of other fabrication methods has been added, and the advantages of electron beam exposure are shown, and the improvements made in this paper on the basis of traditional electron beam lithography fabrication methods are proposed.

  1. Please write what is the Au thickness in 2.1.1. From Fig. 2b it would appear that it is about 1 nm, however no label with units is given with the z-axis, so it is just my guess because the remaining two axes’ units are nm.

A: These comments have been improved in 2.1.1.

  1. The whole grating transfer procedure that you described appears overly complicated and extremely slow. Did you consider an approach that would include the most part of the processing you described, including depositing the gold islands onto a thin polished Si substrate and using a similar functionalization and transfer procedure to the one you already described to "glue" the islands to a new PDMS substrate? The difference is that you could try to etch away the complete silicon substrate in e.g. KOH. It would have lasted much shorter than the proposed removal of sacrificial chromium. In addition, the non-use of chromium would help in decreasing binding between gold cuboids and silicon.

A: There are two reasons for choosing to use a chromium sacrificial layer. First of all, chromium is very conductive and can be used as a conductive layer during electron beam etching. Secondly, no bubbles are generated during the etching of chromium, which is crucial to improve the transfer rate of the structure in this experiment. Moreover, the selectivity of the Cr etching acid solution is very good, and other materials other than chromium will not be corroded during the etching process. But the reviewers raised a good question, slow etch rates are indeed an important issue, and we'd love to try faster etch methods in future work.

  1. The description of coupling between the propagating light beam and the localized surface plasmon polariton as written in lines 157-8 is unclearly written, because it could be understood that a propagating wave could couple with a flat conductive surface without any coupling structure, which is of course impossible. Coupling in your case occurs via diffraction on nanoparticles (and via grating coupling) which adds the necessary additional wavevector. You could rephrase your sentence like “When light illuminates the surface of nanoparticles made from noble metal (or generally from any plasmonic material), the free electrons close to the nanoparticle surface will collectively oscillate.”

A: It’s a very good suggestion, already improved in Section 2.1.3 in the current version.

  1. 8. In Section 4 a comparison of optical images of the structure in Figs 5c and 5d clearly shows that the shape of the flat golden cubelets is elongated along the horizontal axis when the substrate is stretched. Yet, when describing the results in Fig. 6 you comment that there is “no significant change in the shape of the gold structure in the direction parallel to the stretching”. Also you claim that the “vertical length” (is it the thickness or the lateral dimension normal to the stretching direction?) of “cubes” (you do not use any cubes but flat square islands that are best described as “thin cubelets”) does not significantly change during stretching. How is that even possible? Upon stretching, the dimensions along the unstretched axes should be shortened proportionally to the elongation to keep the volume constant. If your explanation were that the shape of the cubelets did not change upon stretching, it would contradict Fig. d. Besides, this would create a huge stress between the stretched substrate and the cubelets, with a result that the interfacial forces would be so large that at least some of the cubelets would simply detach. Kindly elucidate.

A: Thank you for your suggestion, the cube has been changed to thin cubelet. it should be noted that the elongated square structure in Figure 5c and 5d is not a single thin cubelet but a 2D grating consisting of thousands of nano-sized cubelets. the elongation is mainly due to the change of the grating period. A scale bar has been added to the images. The problem of the structure falling off after stretching has been explained in the following questions.

  1. Do repeated stretching/releasing cycles cause detachment/falling off of the gold squares?

A: This is a very critical issue. The work at this stage did not test the stretching limit of the samples, but no significant structural detachment was found in the several stretching experiments that have been performed. However, structural detachment has occurred on samples that have not been chemically functionalized. Even without stretching, the phenomenon of falling off was generated. To solve this problem, we tried wet transfer, where liquid PDMS is poured onto the sample, then cured, and then etched to embed the structure in PDMS. In this case, the stretching would significantly changes the shape of the thin cubelet. Dark field microscopy images after stretching and comparison of spectra before and after stretching have been added in Figure 7.

  1. I had several issues with Fig. 6a. The ordinate axis is labeled by “intensity”. Why is this "Intensity" and not "reflectance"? Why did you use arbitrary units instead of the absolute ones and what do they actually represent? Also, I was unable to distinguish the peak at approx. 530 nm, especially at higher levels of stress, while at the bottommost point I could not discern it from the middle peak. Thus the positions of the dashed lines appear to me somewhat arbitrarily drawn. Your kind explanation would be most welcome here.

A: In this work, the spectra are not obtained from direct measurements. Instead, it is obtained by calibration of four sets of data. As shown in Equation 5. The values of its y-axis have only relative significance. The dashed line is indeed drawn somewhat arbitrarily, but the main purpose is to show the shift trend of the peak. The data in Figure b are from the Lorentz fitting of the spectra with more accurate peak positions.

  1. In the same figure, something is wrong with the color code of the dashed lines. Their order is different between the spectral dependences and stretching percentages. I would expect them to be in identical order.

A: This issue has been corrected in Figure 5

  1. Conclusion seems incomplete to me. What are the prospects for your further work? What are the possibilities of mass fabrication (which are critical for any practical use of your stretchable gratings), especially from the point of view of the fact that the sacrificial chromium removal lasts impractically long? What are the envisioned applications of the structures?

A: A modification has been made to the conclusion section.

  1. As mentioned above, English in the manuscript is rather good. However, that is not to say that it does not need corrections. In the following few points, kindly find my suggestions for language and style improvements.

A: Thank you for your kind comments, have made language adjustments throughout the text.

  1. Please, re-read your manuscript once again and correct the errors like “photonic devices is”, “2.1 theroy”, “in 150 °C”, “under nitrogen”, etc. What is “figure of breath”? “Desired level of PMMA thickness” should be “The desired PMMA thickness.” “The exposure part” can be simply “Exposure.” The sentence “Place 3 ml of HMDS into the crucible in the desiccator and place the holder with the mold on top of the crucible” is unclear, please rephrase it. I believe “Fix both sides of the flexible substrate on the stretching device” should be “Fix two opposing sides of the flexible substrate to the stretching device”

A: The above problems have been corrected in the article

  1. Take special care that the tenses throughout your paper (or at least throughout a single paragraph) match. You cannot start a paragraph in one tense, switch the next sentence to another, then continue in third, it is considered to be bad style. Especially take care that you sometimes needlessly used future tense (e.g. lines 17, 18, or 90) or imperative.

A: Thank you for your comments, improvements have been made

  1. Please don’t use imperative at all (for instance, remove it from the lines 173-177) because you make your procedure description sound like a manual. Use past tense instead, like in the rest of that paragraph.

A: Improvements have been made in section 2.2.

  1. Remove “Track Changes” (red font) from line 103.

A: It has been removed in the current version.

  1. References are well chosen, most of them are quite recent and appropriate. However, it appears to be necessary to add new ones in Introduction, related to the state of the art of the fabrication of nanooptical elements on stretchable substrates.

A: The suggestion has been refined in the introduction section.

  1. In all references the doi numbers are missing.

A: All Doi have been added in the Reference.

Round 2

Reviewer 3 Report

The author responded in all issues, and i recommend to accept the manuscript in the present form.

Author Response

The authors are grateful to the reviewers for their comments on this paper

Reviewer 4 Report

In this major revision of the manuscript the authors accepted a majority of the reviewer’s suggestions, which resulted in a vastly improved text. Only some mostly technical issues have remained, which I think the authors can correct in a simple and straightforward manner. I believe this new version of the paper will merit acceptance after minor revisions, mainly connected with oversights when correcting the previous version.

1. The corrected new version of Abstract still does not contain a short description of the plasmonic metasurface geometry, a sentence mentioning the concrete materials and quantitative data (structure dimensions). A sentence mentioning thin gold cubelets as the meta-atoms of the metasurfaces could have been added, preferably defining all three dimensions of the cubelets as well. This is a simple correction, and I believe it would mean a lot to the readers.

2. There are errors with the references cited in lines 47, 48 (two messages “Error! Reference source not found” appears instead).

3. The flow of the text is seriously interrupted between the newly introduced paragraph in the lines 49-60 and the next one, in the lines 67-74. The first paragraph ends with quoting concrete “critical problems with the fabrication process described above”, while the next one starts with “This has greatly improved the accuracy of the structural shape”. Since the critical problems obviously did not help to greatly improve the accuracy, I conclude that the authors failed to introduce another short paragraph between the two, which would explain what novel methods the authors used to overcome the problems described in the first paragraph. This new paragraph should shortly mention the proximity effect correction (and what is the novelty compared to the countless other works using and the approach to the improvement of adhesion compared to the use of van der Waals forces alone. This correction is simple, but crucial, since it reiterates the main novelty and contribution of your approach.

4. In 2.1.1 I still cannot find the thickness of the gold cubelets, although the authors claim that they introduced that info. Is it equal to the thickness quoted in Section 3, where a value of 50 nm is written?

5. No elaboration on why the authors did not completely etched away the whole Si substrate (fast method) instead of just the chromium sacrificial layer (extremely slow). Is it because of the appearance of hydrogen bubbles during Si etching that could impair the accuracy of the process?

6. The elaboration of Fig. 6 is still lacking important data. The simplest solution to this problem is to incorporate into its explanation most of their own reply to my original suggestion No. 10. Also, the color codes in Fig. 6a are NOT corrected, despite the fact that the authors wrote in their reply that they implemented that correction “in Fig. 5” (which did not need any corrections in the first place and is not related with the described problem whatsoever). Please, do correct the color codes in Fig. 6a so that they are consistent between the diagram lines and the legend in the top right inset of the same figure.

7. Regarding the problem of repeated stretching/releasing cycles causing detachment/falling off of the gold cubelets, you may wish to insert somewhere your own sentence from Reply to reviewer (point 9): “The work at this stage did not test the stretching limit of the samples, but no significant structural detachment was found in the several stretching experiments that have been performed.”

Author Response

Please receive the authors' thanks for your careful reading and detailed comments

 Here are the responses to your comments

1.The corrected new version of Abstract still does not contain a short description of the plasmonic metasurface geometry, a sentence mentioning the concrete materials and quantitative data (structure dimensions). A sentence mentioning thin gold cubelets as the meta-atoms of the metasurfaces could have been added, preferably defining all three dimensions of the cubelets as well. This is a simple correction, and I believe it would mean a lot to the readers.

A: The material of the plasmonic metasurface and the dimensions of the structure in this work have been described in the abstract section.

  1. There are errors with the references cited in lines 47, 48 (two messages “Error! Reference source not found” appears instead).

A:Two citation errors have been fixed [18] [19]

  1. The flow of the text is seriously interrupted between the newly introduced paragraph in the lines 49-60 and the next one, in the lines 67-74. The first paragraph ends with quoting concrete “critical problems with the fabrication process described above”, while the next one starts with “This has greatly improved the accuracy of the structural shape”. Since the critical problems obviously did not help to greatly improve the accuracy, I conclude that the authors failed to introduce another short paragraph between the two, which would explain what novel methods the authors used to overcome the problems described in the first paragraph. This new paragraph should shortly mention the proximity effect correction (and what is the novelty compared to the countless other works using and the approach to the improvement of adhesion compared to the use of van der Waals forces alone. This correction is simple, but crucial, since it reiterates the main novelty and contribution of your approach.

A: This part of the text has been modified by inserting the following text:

“This work used a proximity effect correction method to accurately calculate the exposure dose required for each exposure point. This has greatly improved the accuracy of the structural shape without increasing the experimental complexity.”

  1. In 2.1.1 I still cannot find the thickness of the gold cubelets, although the authors claim that they introduced that info. Is it equal to the thickness quoted in Section 3, where a value of 50 nm is written?

A:The thickness of cubelets is 50nm and has been added in 2.1.1

  1. No elaboration on why the authors did not completely etched away the whole Si substrate (fast method) instead of just the chromium sacrificial layer (extremely slow). Is it because of the appearance of hydrogen bubbles during Si etching that could impair the accuracy of the process?

A:  Firstly, etching silicon with KOH will produce hydrogen bubbles, which will affect the experimental accuracy. In addition, although this work is not experimentally verified, theoretically both strong acid and strong base will cause the breakage of silicon oxygen bond and corrode PDMS. This is the reason why strong base is not used to etch silicon in this work, and the reason was not stated in the paper in order to avoid too lengthy narrative and lack of experimental confirmation of the theory. But it can be used as an attempt for future experiments

  1. The elaboration of Fig. 6 is still lacking important data. The simplest solution to this problem is to incorporate into its explanation most of their own reply to my original suggestion No. 10. Also, the color codes in Fig. 6a are NOT corrected, despite the fact that the authors wrote in their reply that they implemented that correction “in Fig. 5” (which did not need any corrections in the first place and is not related with the described problem whatsoever). Please, do correct the color codes in Fig. 6a so that they are consistent between the diagram lines and the legend in the top right inset of the same figure.

A: Improvements have been made to this section and the color code in Figure 6a has been corrected

  1. Regarding the problem of repeated stretching/releasing cycles causing detachment/falling off of the gold cubelets, you may wish to insert somewhere your own sentence from Reply to reviewer (point 9): “The work at this stage did not test the stretching limit of the samples, but no significant structural detachment was found in the several stretching experiments that have been performed.”

A: The text has been added to the last paragraph of the RESULT section. See text in red
